# Antiparasitic Ovalicin Derivatives from *Pseudallescheria boydii*, a Mutualistic Fungus of French Guiana Termites

**DOI:** 10.3390/molecules27041182

**Published:** 2022-02-10

**Authors:** Jonathan Sorres, Téo Hebra, Nicolas Elie, Charlotte Leman-Loubière, Tatyana Grayfer, Philippe Grellier, David Touboul, Didier Stien, Véronique Eparvier

**Affiliations:** 1CNRS, Institut de Chimie des Substances Naturelles, UPR 2301, Université Paris-Saclay, 91198 Gif-sur-Yvette, France; teo.hebra@cnrs.fr (T.H.); nicolas.elie@cnrs.fr (N.E.); charlotte.leman-loubiere@clarins.com (C.L.-L.); tgrayfer@gmail.com (T.G.); david.touboul@cnrs.fr (D.T.); 2CNRS, Département RDDM, UMR 7245, Muséum National d’Histoire Naturelle CP52, 57 Rue Cuvier, 75005 Paris, France; philippe.grellier@mnhn.fr; 3Laboratoire de Biodiversité et Biotechnologie Microbiennes, CNRS, Sorbonne Université, Observatoire Océanologique, 66650 Banyuls-sur-Mer, France; didier.stien@cnrs.fr

**Keywords:** *Pseudallescheria boydii*, ovalicin, molecular networking, symbiotic fungus, genome, biosynthesis

## Abstract

Social insects are in mutualism with microorganisms, contributing to their resistance against infectious diseases. The fungus *Pseudallescheria boydii* SNB-CN85 isolated from termites produces ovalicin derivatives resulting from the esterification of the less hindered site of the ovalicin epoxide by long-chain fatty acids. Their structures were elucidated using spectroscopic analysis and semisynthesis from ovalicin. For ovalicin, these compounds displayed antiprotozoal activities against *Plasmodium falciparum* and *Trypanosoma brucei,* with IC_50_ values of 19.8 and 1.1 µM, respectively, for the most active compound, i.e., ovalicin linoleate. In parallel, metabolomic profiling of a collection of *P. boydii* strains associated with termites made it possible to highlight this class of compounds together with tyroscherin derivatives in all strains. Finally, the complete genome of *P. boydii* strains was obtained by sequencing, and the cluster of potential ovalicin and ovalicin biosynthesis genes was annotated. Through these metabolomic and genomic analyses, a new ovalicin derivative named boyden C, in which the 6-membered ring of ovalicin was opened by oxidative cleavage, was isolated and structurally characterized.

## 1. Introduction

Insects might be the most represented and diverse living creatures on earth [1]. In nature, insects live in symbiosis with bacteria and fungi to protect themselves from pathogens [2]. These microorganisms play a key role in the ecological success of their host [3], providing them antipathogenic metabolites that could eventually find applications in therapeutics [4]. Insect/microorganism associations are well described for social insects, such as ants, bees, and some wasp species [5,6]. However, little work has been conducted thus far on termite/microorganism interactions (outside the context of trophobioses) [7,8,9,10].

Termites (Blattodea: Isoptera) are eusocial insects comprising more than 3000 species, mostly distributed in tropical and subtropical regions. All termites live in nests containing 1 to 1.5 million individuals divided into castes (reproductive individuals, workers, and soldiers) that cooperatively perform colony tasks. These insects play an important role in ecosystems through the decomposition of plant lignocellulose, having a positive impact in nutrient cycling and soil–water dynamics through their nest construction and tunnel networks in the soil [11]. Termites are vital members of old-growth tropical forests, being perhaps the main decomposers of dead plant material at all stages of humification (decay). Therefore, termite colonies viability is considered a marker of human influence on the environment [12,13]. Nests protect termites from aggressions of biotic and abiotic origin and provide a place for feeding, reproduction, and nursery [14]. Due to their lifestyle, termites developed a singularity related to infection management. Indeed, these social insects live in confined spaces, with a high population density, frequent interactions, genetic homogeneity, and a homeostatic environment. All these factors should favor the transmission of infectious agents, especially pathogenic fungi or bacteria [15]. In addition, termites control the temperature (about 30 °C) and humidity (on average 90%) of their nests, in order to raise immatures, inducing favorable conditions for the development of pathogens but without any alteration of the viability of termite colonies [16,17].

The investigation of under-explored microbial niches, such as termite-associated microorganisms, can provide original active metabolites [4,7,8,9,10]. It should also be mentioned that microorganisms constitute one of the most important forms of life that provide biotechnological tools for producing highly valuable chemicals [18], making the investigation of unexplored microbes one the more important avenues for paving future discoveries in natural product research [19].

Previous studies in our group led to the isolation of six fungal strains of the genus *Pseudallescheria* from three different termite colonies (out of eight nests harvested) [20,21,22]. Our preliminary results suggest that the fungi genus *Pseudallescheria* may be frequently associated with termites, exchanged between individuals, and spread in the nest. It was first demonstrated that these fungi produce antifungal compounds, in particular. It was hypothesized that this fungus has the ability to protect termites, its macroscopic host. *Pseudallescheria boydii* is a fungus frequently isolated from insects [20,21,22,23]. It seems that *P. boydii* has a broad host range, mainly described as saprophytic, frequently present in soil or polluted water but also on the surface of seaweed or endophytic [24,25] and in humans by causing invasive infections mainly in immunocompromised hosts but also in immunocompetent hosts [26]. The anamorphic form of *P. boydii* is *Scedosporium apiospermum*. It has been described that these strains can produce a vast diversity of alkaloids, cyclopeptides, and polyketides. Some of these compounds, such as boydens and gliotoxins, have shown antimicrobial activity [20,21,22,23,24,25,26,27,28]. In our collection, it was shown that two *P. boydii* strains (SNB-CN85 and SNB-CN73) produce tyroscherine, ovalicin (two antifungal compounds), and analogs of these two specialized metabolites [20,21,22].

In parallel, new methodologies for dereplication of complex mixtures have been developed to accelerate and expand the discovery of new bioactive natural molecules. In order to avoid focusing research efforts on already known molecules, dereplication strategies have been developed [29,30]. Dereplication allows for the annotation of known molecules in a mixture but also allows for targeting and focusing on potentially novel molecules to be isolated and characterized. Mass spectrometry-based molecular networking approaches can be used to overcome these difficulties, allowing the targeting of specific metabolites through the tandem mass spectrometry data set organization they confer [29,30,31,32,33,34,35]. In this study, a molecular network approach allowed us to navigate the chemical space of *P. boydii* metabolomes and was used to annotate the chemical diversity of these fungi. A new compound was identified by this method and was isolated and fully characterized. Then, it was chosen to compare the metabolomic and genomic data of all *P. boydii* in our collection. For this purpose, the whole genome sequence of the four *P. boydii* strains was established, and the cluster of potential genes for biosynthesis of ovalicin and its fatty acid derivatives was annotated.

## 2. Results

The study of the *P. boydii* SNB-CN85 ethyl acetate extract led to the isolation of ovalicin (**1**) and the five ovalicin analogs **2**–**6** (Figure 1).

Preliminary inspection of the ^1^H and ^13^C NMR spectra confirmed that Compounds **2**–**5** were ovalicin analogs (Appendix A, Table 1, Appendix A). In fact, Compounds **2**–**5** exhibited the same ^1^H and ^13^C NMR shifts as ovalicin, and the only difference was observed at position 14 with protons and carbons at δ_H_ 4.11 and 4.17/δ_C_ 69.4–69.5 ppm instead of δ_H_ 2.63 and 2.97/δ_C_ 51.0 ppm for ovalicin. For Compound **6**, the same chemical shift but also the loss of the *O*-methyl group at position 7 was observed (Appendix A).

The molecular formula of Compound **2** was determined to be C_34_H_56_O_7_ based on the sodiated molecular ion signal at *m/z* 599.3931 [M + Na]^+^ in the ESI-HRMS analysis (calcd. for C_34_H_56_O_7_Na, 599.3924, err 1.7 ppm). Examination of the ^1^H, ^13^C, HSQC spectra and ^1^H-^1^H COSY and ^1^H-^13^C HMBC correlations indicated the presence of an octadecadienoate chain, as shown by the presence of four olefinic methines (4H at δ_H_ 5.33–5.36), 12 methylenes (δ_H_ 1.28–2.78), and one triplet methyl (δ_H_ 0.91). The chain was also determined following the sequence of ^1^H-^1^H COSY cross-peaks H-8′/H-9′/H-10′/H-11′/H-12′/H-13′. The fatty acid chain was linked to C-14 thanks to the H-14/C-1′ correlation (Figure 2). Compounds **3**–**4** and **6** presented similar spectral data. The only difference was observed in the fatty acid chains. The exact mass of each molecule, the presence of olefinic protons between δ_H_ 5.33 and 5.36 ppm, and the ^1^H-^1^H COSY correlations confirmed the presence of two double bonds for **2** and **6**, one double bond for **3** and **5**, and none for **4** (Figure 2).

Compounds **2**–**4** and **6** possess an 18-carbon fatty acid chain, while **5** has a shorter 16-carbon chain. The positions of the double bonds of natural Compounds **2** and **3** were determined by MS/MS after hydrolysis of the fatty chains and fragmentation of in-source acetonitrile adducts (Appendix A) [36,37]. Due to too small quantities, this procedure was not possible for Compounds **5** and **6**. To confirm the structures and configurations of Compounds **2**, **3**, and **4**, ovalicin esters were synthesized from Compound **1** by esterification with linoleic acid, oleic acid, and stearic acid, respectively (Appendix A). The comparison of the NMR spectra obtained between the isolated and synthetic compounds allowed us to confirm the identification and configurations of fungal metabolites **2**–**4** (Appendix A). The absolute configurations Compounds **2**–**4** were also confirmed first by comparison with circular dichroism data of 1 (Appendix A) for the ovalicin moiety. Furthermore, all these compounds showed negative optical rotation. Secondly, for the fatty acid moiety, our ^13^C NMR analyses were in accordance with literature data concerning the specific chemical shifts of methylene at C-11′ position between two double bonds (Appendix A) [38,39].

Compounds **1**–**4** antiplasmodial activity against *Plasmodium falciparum* and antitrypanosomal activity against *Trypanosoma brucei brucei* are summarized in Table 2. Ovalicin **1** and fatty acid ester of ovalicin **2** and **4** showed high inhibition percentages at IC_50_ values below 20.6 µM against *Plasmodium falciparum*. Against *Trypanosoma brucei brucei*, all compounds exhibited IC_50_ values below 5 µM, but **1** to **2** were the only compounds to show activity (IC_50_ < 1.1 µM). Compounds **5** and **6** were isolated in too small amounts and could not be tested. To the best of our knowledge, only ovalicin is known to show antiangiogenic properties [40] and potent antiplasmodial activity [41]. Consequently, the isolated compounds were also tested for their cytotoxicity on normal primary human umbilical vein endothelial cells (HUVECs). All compounds demonstrated low cytotoxicity on the tested cell lines (IC_50_ > 20 µM), contrary to reported literature for ovalicin with an IC_50_ of 13 µM [42,43]. Finally, isolated compounds were also tested for their potential antimicrobial activities against *C. albicans*. The minimal inhibitory concentrations (MICs) observed for molecules **1**–**3** were greater than 128 µg/mL, showing no significant inhibition.

As it has been previously shown that strains of *P. boydii* were able to biosynthesize derivatives of tyrocherin and ovalicin [20,21,22], an evaluation was conducted of whether our different strains were also capable of biosynthesizing the fatty acid derivatives of ovalicin isolated from SNB-CN85. For this purpose, EtOAc extracts of four *P. boydii* strains, SNB-CN71, -CN73, -CN81, and -CN85, were analyzed by reversed-phase liquid chromatography–high-resolution mass spectrometry (RPPLC-HRMS/MS) using the data-dependent acquisition mode. The resulting spectral data were further investigated by the molecular network approach (MN) [29,30,31]. Data were processed by MZmine 2.51 [32,33], and an MN was generated by Metgem [34]. The resulting network grouped 3021 nodes into 83 clusters (Figure 3 and Appendix A). Strain mapping was applied by attributing a given color code to each strain in the collection, allowing us to visually navigate through the MN and check for the presence and distribution of specific compound classes within the collection. Relative quantification of the ions was represented using pie charts, with their proportions based on the respective areas of the peaks detected on extracted ion chromatography (XIC).

Clusters were identified as primarily related to the tyrocherin and pseurotin A analogs (Figure 3). The three known specialized metabolites, i.e., tyrocherin ([M + H]^+^*, m/z* 334.2738, err. 0.8 ppm, C_21_H_35_NO_2_), *N*-methyltyrocherin ([M + H]^+^, *m/z* 348.2872, err. 7.2 ppm, C_22_H_37_NO_2_), and pseurotin A ([M + Na]^+^, *m/z* 454.1438, err. 8 ppm, C_22_H_25_NO_8_), were present in a large relative proportion in SNB-CN71, -CN73, and -CN81. *N*-Methyltyrocherin and pseurotin A were not detected in SNB-CN85 and SNB-CN71, but they were previously isolated from these strains. Therefore, pseudallicins A–D isolated previously from SNB-CN85 extract strains were not found during this analysis. This may be because the concentration of these compounds in the crude extracts appeared to be very low, whereas they had been concentrated in the previously studied fractions. Sodium-cationized ovalicin and Compounds **1**–**5** were detected in a specific cluster present in all extracts. By studying the fragmentation spectra of the compounds grouped in the ovalicin fatty acid cluster, it was possible to determine the presence of ovalicin analogs bearing C18 fatty acid chains with three double bonds (Compound **9**), one to two epoxides (Compounds **10**, **11** and **12**), or with a C16 fatty acid chain (Compounds **5** and **8**). Compound **11** exhibited both an epoxide and a double bond (Appendix A). It is interesting to note that this collection of fatty acid esters of ovalicin is quite large, and some of these specialized metabolites are present in the different species of the collection. Finally, not all strains produced the different derivatives in the same proportions, although they were grown under the same conditions. Compound **6** was not detected in the cluster, likely because this compound is a demethylated derivative of ovalicin but mostly because it seems to be present in very small quantities in the crude extract.

Following these results, the biosynthetic pathway of ovalicin was investigated. For this purpose, whole genome sequencing of the four *P. boydii* strains SNB-CN71, -CN73, -CN81, and -CN85 was performed by combining short and long sequences to achieve de novo hybrid assemblies. The quality of the four 42–43 Mb genomes was assessed by the BUSCO score (95.9–96.1% completeness) (Appendix A) [44]. It was considered that ovalicin is a known fumagillol analog, an intermediate in the fumagillin biosynthetic pathway [45,46]. While the fumagillin biosynthetic pathway has been described [47], this is not the case for ovalicin and its fatty ester derivatives. Furthermore, β-trans-bergamotene is a biosynthetic intermediate that ovalicin shares with fumagillin and fumagillol [45,46]. When our four genomes were analyzed using the antiSMASH pipeline [48], a cluster of biosynthetic genes common to all four strains, comprising enzymes responsible for producing ovalicin and its fatty acid esters, was identified. This cluster, composed of 18 genes conserved in the four *P. boydii* strains, was intertwined with the pseurotin A biosynthetic gene cluster (6 genes out of 18). Pseurotin A is another specialized metabolite produced by *P. boydii* strains whose protonated species were identified in the molecular networks (Figure 3, Appendix A). These two gene clusters appear to be frequently intertwined, as the pseurotin/fumagillin gene cluster has also been reported in *Aspergillus fumigatus* [49]. The eight genes coding for enzymes involved in the ovalicin biosynthetic pathway were named *Pbo_Ova_A*–*H* (Figure 4, Appendix A), and the six genes coding for pseurotin *A* were named *Pbo_Pse_A*–*F* (Appendix A). Their functions were predicted using the Pfam and UniProt databases [50,51]. Like fumagillin, the ovalicin biosynthetic pathway is initiated by the action of the terpene synthase Pbo_Ova_A, which converts farnesyl pyrophosphate to β-trans-bergamoten (Btb). A cytochrome P450 monooxygenase Pbo_Ova_B whose af510 is the closest homolog (85% similarity) was predicted to catalyze the formation of intermediate (**Btb1**) in which protonated species were found in RPLC-ESI(+)–MS/MS data ([M + H]^+^, *m/z* 251.1634, err. 3.1 ppm) (Appendix A). A hydroxyl was added at position 3 by Pbo_Ova_C, a homologous protein of af480 (75% similarity) to form compound **Btb2**, whose protonated species was also found in the RPLC-ESI(+)–MS/MS data ([M + H]^+^, *m/z* 267.1587, err. 1.5 ppm). Then, a methyl group was added by Pbo_Ova_D, a methyl transferase, to the hydroxyl group at position C-7 to form molecule **Btb3** ([M + Na]^+^, *m/z* 303.1564, err. 1.0 ppm). From this step, the ovalicin biosynthetic pathway differed from that of fumagillin: where compound **Btb3** was converted into ovalicin (**1**) ([M + Na]^+^, *m/z* 319.1512, err. 1.3 ppm) by the addition of a hydroxyl at position C-5. This catalysis was likely performed by the Pbo_Ova_E oxidoreductase or Pbo_Ova_E dioxygenase present in the biosynthetic gene cluster. Then, Pbo_Ova_G hydrolase cleaved the position C-4 ester bond of ovalicin (**1**) to form dihydro-ovalicin ([M + Na]^+^, *m/z* 321.1669, err. -1.2 ppm), which shares a neutral loss with Compound **1** equivalent to a cosine score of 0.34 (Appendix A). Finally, fatty acids were likely added to dihydroovalicin via an acyltransferase, the most likely candidate of which was Pbo_Ova_G, whose gene was present at the edge of the Pbo_Ova/Pse biosynthetic gene cluster.

This biosynthetic scheme was fully consistent with experimental data proposed previously, concerning configurations of compounds **2**–**6** and sheds light on subunits’ ovalicin absolute configurations.

To more deeply explore the chemical diversity of the *P. boydii* metabolome, we looked for nodes with intense signals but could not annotate them by searching standards and analogs in MetGem databases. It was chosen to isolate the specialized metabolites showing an ion signal at *m/z* 323.1433 (Figure 3 and Appendix A), which could be a potential original derivative of ovalicin in view of the observed mass. The molecular formula of Compound **7** was determined to be C_15_H_24_O_6_ by ESI-HRMS (*m/z* 323.1473 [M + Na]^+^, calcd. for C_15_H_24_O_6_Na^+^, 323.1471, err 1.3 ppm), indicating the presence of two additional oxygen and the lack of one carbon compared with the molecular formula of ovalicin (**1**). The ^1^H NMR spectra of Compounds **1** and **7** displayed some similarities, suggesting that **7** was an analog of **1**. The remarkable difference between Compounds **1** and **7** in ^1^H NMR was the absence, in **7**, of the singlet methoxy signal integrating for 3H at δ_H_ 3.49 (Table 3).

However, Compound **7** exhibited characteristic dimethylallyl signals at δ_H_ 5.24 (m, 1H), δ_H_ 1.73 (m, 3H), and 1.67 (m, 3H). The examination of the COSY correlations allowed us to link this dimethylallyl moiety to H-8 at 3.26, (t, *J* = 6.5 Hz, 1H) [8]. The complete proton assignment of Compound **7** was deduced from the careful examination of COSY, HSQC, and HMBC correlations (Figure 5). HMBC analysis indicated that methylene protons at position 2 at δ_H_ 2.22 ppm correlated with a carbonyl carbon of a carboxylic acid at δ_C_ 178.0 ppm. The analysis of the HSQC correlations showed that Compound **7** had additional oxygenated methylene proton signals at δ_H_ 3.64 and 3.60 ppm. Consequently, we deduced that Compound **7** was an analog of demethylated ovalicin in which the C-1–C-7 bond was cleaved by oxidation or hydrolysis. The relative and absolute stereochemistry at C-7 was defined as drawn in Figure 5 based on biosynthetic considerations (Figure 4). Compound **7** was named boyden C. Unfortunately, as this compound was isolated in small quantities, it could not be biologically tested.

## 3. Materials and Methods

### 3.1. General Experimental Procedures

Optical rotations ([α]D) were measured on an Anton Paar MCP 300 polarimeter in a 100 mm long 350 μL cell (Anton Paar, Graz, Austria). IR spectra were recorded using a Perkin Elmer Spectrometer BX FT-IR (Thermo Fisher scientific, Les Ulis, France). Nuclear magnetic resonance (NMR) spectra were recorded on a Bruker 600 MHz spectrometer equipped with a 3 mm inverse detection cryoprobe or a Bruker 500 and 800 MHz spectrometer equipped with a 5 mm inverse detection probe (Bruker, Rheinstetten, Germany). Chemical shifts (δ) are reported as ppm based on the TMS signal. HR-ESI-MS measurements were performed using a Waters Acquity UPLC system with column bypass coupled to a Waters LCT Premier time-of-flight mass spectrometer equipped with an electrospray interface (ESI) (Waters, Manchester, England). Flash chromatography was performed on a Grace Reveleris system with dual UV and ELSD detection equipped with a 40 g C18 column. Analytical and preparative HPLCs were conducted with a Gilson system equipped with a 322 pump device, a GX-271 fraction collector, a 171 diode array detector, and a prepELSII electrospray nebulizer detector (Gilson, Middelton, WI, USA). Columns used for these experiments included a Phenomenex Luna C18 5 μm 4.6 × 250 mm analytical column and a Phenomenex Luna C18 5 μm 21.2 × 250 mm preparative column. The flow rate was set to 1 or 21 mL/min using a linear gradient of H_2_O mixed with an increasing proportion of CH_3_CN. Both solvents were HPLC grade and modified with 0.1% formic acid. Potato dextrose agar (PDA) was purchased from Fluka Analytical (Janssen Pharmaceuticalaan, Geel, Belgium). Molecular analyses were performed externally by BACTUP, Saint-Priest, France. MS/MS experiments were performed by direct injection analysis (DIA, 10 μL/min) on an Agilent Q-ToF 6540 (APCI interface) mass spectrometer (Agilent, Santa Clara, CA, USA). Compounds were dissolved in CH_2_Cl_2_ at 2 mg/mL and then diluted in CH_3_CN at 200 μg/mL. The corona current was fixed to 2 µA, while the desolvation gas flow was set to 8 L/min (heated at 300 °C). The capillary, fragmentor, and skimmer voltages were set to 3000 V, 100 V, and 45 V, respectively. A collision energy of 15 eV was used.

### 3.2. Collection and Identification of Pseudallescheria boydii

Termite workers were collected from an aerial termite nest located in Rémire-Montjoly, French Guiana, in July 2011. The termites were identified as *Termes* cf. *hispaniolae* and *Nasutitermes corniger* by Prof. Reginaldo Constantino (University of Brasilia, Brazil). The workers were surface-sterilized by successive soakings in 70% EtOH (2 min), 5% NaOCl (2 min), and sterile water. The termite was subsequently placed in a Petri dish containing solid PDA medium. After 1 week at 25 °C, the first fungal hyphae to emerge from the insect were sampled and transferred into other Petri dishes. One of the microbial colonies consisted of a pure fungus, which was saved in triplicate at −80 °C in H_2_O−glycerol (50/50). A sample was submitted for amplification of the nuclear ribosomal internal transcribed spacer region ITS. Sequencing allowed for strain identification by NCBI sequence comparison (Blastn^®^, Rockville, MD, USA). The sequences have been registered in the NCBI GenBank database (http://www.ncbi.nlm.nih.gov, accessed on 20 December 2021) under the registry numbers KJ023743 (SNB-CN85), KC684883 (SNB-CN73), KJ023735 (SNB-CN71), and KJ023720 (SNB-CN81).

### 3.3. Culture, Extraction, and Isolation

The *P. boydii* strains were cultivated on PDA at 26 °C for 15 days, initially on a small scale. The fungus and culture medium were then transferred into a large container and macerated with EtOAc for 24 h. The organic solvent was then collected by filtration, washed with H_2_O in a separatory funnel and evaporated, yielding the four crude extracts.

The *P. boydii* strain SNB-CN85 was cultivated on a large scale in 150 × 14 cm Petri dishes, yielding 3.57 g extract. A portion of the extract (1.13 g) was subjected to reversed-phase flash chromatography using a gradient of H_2_O mixed with an increasing proportion of CH_3_CN to afford 6 fractions (A to E). The column was then eluted with CH_3_CN/CH_2_Cl_2_ 1:1 (Fraction F). Fraction D (300 mg, eluted with H_2_O/CH_3_CN, 2:8) was purified by reversed-phase flash chromatography using the same solvent system as mentioned above with a modified gradient. The major compound was collected in pure form and was identified as ovalicin (**1**) (28.1 mg). Fraction F (89 mg) was subjected to preparative HPLC (Luna C_18_, elution gradient from 10:90 to 0:100 over 30 min of H_2_O/ACN) to afford ovalicin linoleate (**2**) (2.2 mg, RT = 23.0 min), ovalicin oleate (**3**) (1.5 mg, RT = 30.2 min), fraction *i* (RT = 22.3–22.5 min), and fraction *ii* (RT = 24.8–30.1 min). Fraction *i* was subjected to preparative HPLC (Luna PFP(2), H_2_O/ACN 25:75) to afford demethylovalicin linoleate (**6**) (<0.1 mg, RT = 16.1 min) and ovalicin palmitoleate (**5**) (<0.1 mg, RT = 17.1 min). Fraction *ii* was subjected to preparative HPLC (Luna PFP(2), elution gradient from 20:80 to 15:85 over 30 min) to afford ovalicin stearate (**4**) (1.0 mg, RT = 15.0 min).

*P. boydii* SNB-CN73 was also cultivated on a large scale, similar to SNB-CN85. It yielded 2.46 g of extract. A portion of the extract (1.23 g) was purified by flash chromatography with a linear gradient of hexane−EtOAc followed by another gradient of EtOAc−MeOH. Twelve fractions were gathered based on their TLC profiles. Fraction 11 (29.6 mg), which contained Compound **7,** was subjected to preparative HPLC (Luna PFP, elution gradient H_2_O/ACN from 35:65 to 0:100 over 30 min) to afford **7** (1.1 mg, RT = 6.0 min).

*Ovalicin* (**1**), white powder, [α]^20^_D_ −55.7 (c 0.14, MeOH)/−77.7 (c 0.13 CDCl_3_), ^1^H NMR (500 MHz, CDCl_3_) (Appendix A), ESI-HRMS *m/z* 297.1702 [M + H]^+^ (calcd. for C_16_H_25_O_5_^+^, 297.1702) (Appendix A).

*Ovalicin linoleate* (**2**), white amorphous solid, [α]^20^_D_ −26.2 (c 0.4, MeOH), ^1^H and ^13^C NMR (Appendix A), ESI-HRMS **2**
*m/z* 599.3931 [M + Na]^+^ (calcd. for C_34_H_56_O_7_Na^+^, 599.3924) (Appendix A).

*Ovalicin oleate* (**3**), colorless amorphous solid, [α]^20^_D_ −15.7 (c 0.1, MeOH), ^1^H and ^13^C NMR (Appendix A), ESI-HRMS **3**
*m/z* 601.4091 [M + Na]^+^ (calcd. for C_34_H_58_O_7_Na^+^, 601.4080) (Appendix A).

*Ovalicin stearate* (**4**), colorless amorphous solid, [α]^20^_D_ −20.4 (c 0.1, MeOH), ^1^H and ^13^C NMR (Appendix A), ESI-HRMS *m/z* 603.4223 [M + Na]^+^ (calcd. for C_34_H_60_O_7_Na^+^, 603.4237) (Appendix A).

*Ovalicin palmitoleate* (**5**), colorless amorphous solid, [α]^20^_D_ −21.7 (c 0.1, MeOH), ^1^H NMR, (Appendix A), ESI-HRMS *m/z* 573.3793 [M + Na]^+^ (calcd. for C_32_H_54_O_7_Na^+^, 573.3767) (Appendix A).

*Demethylovalicin linoleate* (**6**), colorless amorphous solid, [α]^20^_D_ −30.2 (c 0.1, MeOH), ^1^H and ^13^C NMR, (Appendix A), ESI-HRMS *m/z* 585.3782 [M + Na]^+^ (calcd. for C_33_H_54_O_7_Na^+^, 585.3767) (Appendix A).

*Boyden C* (**7**), white powder, [α]^20^_D_ −10.0 (c 0.1, MeOH), ^1^H and ^13^C NMR, (Appendix A, Table 2, Appendix A); ESI-HRMS [M + H]^+^ *m/z* 301.1654 (calcd. for C_15_H_25_O_6_^+^ *m/z* 301.1651) (Appendix A).

### 3.4. Fatty Acid Methyl Ester Synthesis from Ovalicin Linoleate (***2***) and Ovalicin Oleate (***3***)

#### 3.4.1. Hydrolysis

Compound **2** (0.4 mg, 0.69 μmol) and Compound **3** (0.4 mg, 0.69 μmol) were each dissolved in a 1:4 MeOH/THF solution (500 μL). LiOH (0.6 mg, 25.0 μmol) and distilled water (50 μL) were added, and the mixture was stirred for 4 h at 60 °C. The solution was concentrated under reduced pressure, and the residue was dissolved in distilled water (700 μL). After the addition of 0.1 M HCl (300 μL), the aqueous layer was extracted with distilled heptane (3 × 0.5 mL). The combined organic layers were filtered on Na_2_SO_4_ and concentrated to afford free fatty acids.

#### 3.4.2. Esterification

The free fatty acids were dissolved in MeOH (500 μL). Each solution was cooled down to 0 °C. SOCl_2_ (20 μL, 275.4 μmol) was added, and the solution was stirred for 1 h at 50 °C. The solution was concentrated under reduced pressure, and H_2_O (500 μL) was added. The aqueous solution was extracted with a mixture of Et_2_O/heptane, 5:95 (2 mL). The organic layers were filtered on a Na_2_SO_4_/SiO_2_ column and concentrated to afford fatty acid methyl esters.

### 3.5. Semisynthesis of Ovalicin Linoleate, Ovalicin Oleate, and Ovalicin Stearate

#### 3.5.1. Synthesis of Ovalicin Linoleate (**2b**)

Ovalicin (**1**, 17.9 mg, 0.06 mmol), linoleic acid (20.4 mg, 0.072 mmol), and KOH (0.7 mg, 0.012 mmol) were dissolved in anhydrous DMSO (0.5 mL). The mixture was stirred for 72 h at 60 °C. The solution was concentrated under reduced pressure, and the residue was dissolved in distilled water (1.5 μL). After addition of 0.1 M HCl (0.1 mL), the aqueous solution was extracted with Et_2_O (2 × 2.0 mL). The combined organic layers were washed with water (2 mL), filtered, and evaporated to afford a crude mixture (32.2 mg), which was subjected to preparative HPLC (Luna C18, isocratic 5:95 elution over 20 min) to afford synthetic ovalicin linoleate (**2**) (12.1 mg, RT = 12.8 min).

*Synthetic ovalicin linoleate* (**2b**): Yield: 35%, white amorphous solid, [α]^22^_D_ −28.7 (c 0.1, MeOH), ^1^H NMR spectrum identical to that of natural **2**, *m/z* 599.3911 [M + Na]^+^ (calcd. for C_34_H_56_O_7_Na^+^, 599.3924).

#### 3.5.2. Synthesis of Ovalicin Oleate (**3b**)

Compound **3** was synthetized according to the procedure mentioned above, with **1** (15.3 mg, 0.052 mmol), oleic acid (17.8 mg, 0.072 mmol), and KOH (0.7 mg, 0.012 mmol). The crude mixture obtained (19.6 mg) was subjected to preparative HPLC to afford synthetic ovalicin oleate (**3**) (6.5 mg, RT = 18.2 min).

*Synthetic ovalicin oleate* (**3b**): Yield: 22%, white amorphous solid, [α]^22^_D_ −22.4 (c 0.1, MeOH), ^1^H NMR spectrum identical to that of natural **3**, *m/z* 601.4056 [M + Na]^+^ (calcd. for C_34_H_56_O_7_Na^+^, 601.4080).

#### 3.5.3. Synthesis of Ovalicin Stearate (**4b**)

Compound **4** was synthetized according to the procedure mentioned above with **1** (14.7 mg, 0.050 mmol), stearic acid (21.3 mg, 0.074 mmol), and KOH (1 mg, 0.018 mmol). The crude mixture obtained (31.0 mg) was subjected to preparative HPLC (Luna C18, isocratic 0:100 elution over 20 min) to afford synthetic ovalicin stearate (**4**) (6.5 mg, RT = 18.0 min).

*Synthetic ovalicin stearate* (**4b**): Yield: 22%, white amorphous solid, [α]^22^_D_ −17.0 (c 0.1, MeOH), ^1^H NMR spectrum identical to that of natural Compound **4**, ESI-HRMS *m/z* 603.4229 [M + Na]^+^ (calcd. for C_34_H_60_O_7_Na^+^, 603.4237).

### 3.6. Identification of the Double-Bond Position in Fatty Acid Methyl Esters by Liquid Chromatography/Atmospheric Pressure Chemical Ionization Mass Spectrometry

The double-bond positions in the fatty acid methyl esters synthetized from ovalicin linoleate (**2**) and ovalicin oleate (**3**) were determined using the method developed by Vrkoslav et al. (Appendix A) [36,52].

### 3.7. Metabolomic Profiling and Molecular Network Analysis

#### 3.7.1. LC–MS/MS and Data Analysis

LC–MS/MS experiments were performed with a 1260 Prime HPLC (Agilent Technologies, Waldbronn, Germany) coupled with an Agilent 6540 Q-ToF (Agilent Technologies, Waldbronn, Germany) tandem mass spectrometer. LC separation was achieved with an Accucore RP-MS column (100 × 2.1 mm, 2.6 μm, Thermo Scientific, Les Ulis, France) with a mobile phase consisting of H_2_O/formic acid (99.9/0.1) (A)–acetonitrile/formic acid (99.9/0.1) (B). The column oven was set at 45 °C. Compounds were eluted at a flow rate of 0.4 mL·min^−1^ with a gradient from 5% B to 100% B in 20 min and then 100% B for 3 min. The injection volume was fixed at 5 μL for all analyses. Mass spectra were recorded in electrospray ionization positive ion mode with the following parameters: gas temperature 325 °C, drying gas flow rate 10 L·min^−1^, nebulizer pressure 30 psi, sheath gas temperature 350 °C, sheath gas flow rate 10 L·min^−1^, capillary voltage 3500 V, nozzle voltage 500 V, fragmentor voltage 130 V, skimmer voltage 45 V, Octopole 1 RF Voltage 750 V. Internal calibration was achieved with two calibrants, purine and hexakis (1H,1H,3H–tetrafluoropropoxy) phosphazene (*m/z* 121.0509 and *m/z* 922.0098), providing a high mass accuracy better than 10 ppm. The data-dependent MS/MS events were acquired for the five most intense ions detected by full-scan MS, from the 200–1000 *m/z* range, above an absolute threshold of 1000 counts. Selected precursor ions were fragmented at a fixed collision energy of 30 eV and with an isolation window of 1.3 amu. The mass range of the precursor and fragment ions was set as *m/z* 200–1000.

#### 3.7.2. Data Processing and Analysis

The data files were converted from the .d standard data format (Agilent Technologies, Waldbronn, Germany) to .mzXML format using MSConvert software, part of the ProteoWizard package 3.0 [53]. All mzxml values were processed using MZmine2v51, as previously described [33,34]. Mass detection was realized with an MS1 noise level of 1000 and an MS/MS noise level of 0. The ADAP chromatogram builder was employed with a minimum group size of scans of 3, a group intensity threshold of 1000, a minimum highest intensity of 1000, and *m/z* tolerance of 0.008 (or 20 ppm). Deconvolution was performed with the ADAP wavelet algorithm according to the following settings: S/N threshold = 10, minimum feature height = 1000, coefficient/area threshold = 10, peak duration range 0.01–1.5 min, and *t_R_* wavelet range 0.00–0.04 min. MS/MS scans were paired using a *m/z* tolerance range of 0.05 Da and *t_R_* tolerance range of 0.5 min. Isotopologues were grouped using the isotopic peak grouper algorithm with a *m/z* tolerance of 0.008 (or 20 ppm) and a *t_R_* tolerance of 0.2 min. Peaks were filtered using a feature list row filter, keeping only peaks with MS/MS scans (GNPS). Adduct identification, i.e., sodium- or potassium-cationized species, was performed on the peak list with a retention time tolerance of 0.1 min, a *m/z* tolerance of 0.008 or 20 ppm, and a maximum relative peak height of 150%. A complex search, such as dimers, was performed with a retention time tolerance of 0.1 min, a *m/z* tolerance of 0.008 or 20 ppm, and a maximum relative peak height of 150%. Peak alignment was performed using the join aligner with a *m/z* tolerance of 0.008 (or 20 ppm), a weight for *m/z* at 20, a retention time tolerance of 0.2 min, and weight for *t_R_* at 50. The MGF file and the metadata were generated using the export/submit to GNPS option (https://doi.org/10.5281/zenodo.5733173, accessed on 10 January 2019).

Molecular networks were calculated and visualized using MetGem 1.34 software (https://metgem.github.io/, accessed on 1 March 2019) [34], and MS/MS spectra were window-filtered by choosing only the top 6 peaks in the ±50 Da window throughout the spectrum. The data were filtered by removing all peaks in the ±17 Da range around the precursor *m/z*. The *m/z* tolerance windows used to find the matching peaks were set to 0.02 Da, and cosine scores were kept in consideration for spectra sharing at least 2 matching peaks. The number of iterations, perplexity, learning rate, and early exaggeration parameters were set to 5000, 25, 200, and 12 for the t-SNE view.

### 3.8. Genomic Analyses

#### 3.8.1. gDNA Extraction of *P. boydii* Strains SNB-CN71, -CN73, -CN81, and -CN85

For high molecular size gDNA extraction, the strains were cultivated on PDA medium on permeable cellophane membranes that were prepared as described by Fauchery et al. [54]. The cellophane membranes were trimmed to the size of the Petri dish, put into boiling distilled water containing EDTA (1 g·L^−1^) 20 min in order to permeabilize the membrane, rinsed four times in a big container with distilled water, and autoclaved. After 7 days at 26 °C microorganisms were removed from cellophane membrane, snap-frozen, and milled using mortar and pestle. gDNA was extracted from microorganisms using NucleoBond Buffer Set III and AXG 100 (Macherey-Nagel, Hoerdt, France) with slight modifications: 250 mg of milled mycelium was resuspended in 5 mL of buffer G3, 37 °C for 14 h; 1.2 mL of buffer G4 was added; and the solution was gently mixed then incubated at 50 °C for 4 h. Sample was clarified through centrifugation 5000× *g*, 5 min. After ethanol precipitation, samples were re-dissolved in 100 µL of mQ water. Purity control was carried out on a Nanodrop 2000 spectrophotometer, DNA quantity using a Qubit dsDNA BR assay kit according to the manufacture’s recommendation and a Qubit fluorometer, and quality control on 0.7% agarose gel.

#### 3.8.2. Whole Genome Sequencing and Hybrid Assembly

Whole genome sequencing and hybrid assemblies were performed externally by high-throughput sequencing core facility of I2BC (Centre de Recherche de Gif—http://www.i2bc.paris-saclay.fr/, accessed on 8 September 2020). Briefly, purified DNA libraries were loaded on FLO-MIN106 flowcell and long-read DNA sequences were obtained on a GridION (Oxford Nanopore Technologies). The same DNA used for long-read sequencing was fragmented and sequenced using a paired-end strategy on an Illumina NextSeq 500/550. Basecalling and adapter trimming were performed using guppy 4.3.4-1 and Porechop 0.2.4 for long reads. For short reads, demultiplexing, adapter trimming, and quality control were performed using bcl2fastq2-2.18.12, Cutadapt 1.15, and FastQC v0.11.5, respectively. Finally, genome assembly was achieved using Filtlong to filter reads for length and quality (https://github.com/rrwick/Filtlong, accessed on 11 July 2020), Canu for long-reads assembly (https://github.com/marbl/canu, accessed on 11 July 2020) [55], and Pilon to correct assembly by short reads (https://github.com/broadinstitute/pilon, accessed on 11 July 2020) [56]. Finally, the quality of assembly was assessed using BUSCO [57].

### 3.9. Biological Tests

#### 3.9.1. Antimicrobial Assays

Pure isolated compounds were tested on human pathogenic microorganisms such as *Candida albicans* (ATCC 10213) to evaluate their antifungal activities. The test was performed in conformance with a reference protocol from the European Committee on Antimicrobial Susceptibility Testing (EUCAST). The MIC value was obtained after 48 h for *C. albicans*, and fluconazole (for fungi) was used as a positive control [58].

#### 3.9.2. Antiparasitic Activities

##### Assay for In Vitro Inhibition of *T. brucei brucei* Growth

Bloodstream forms of *Trypanosoma brucei brucei* strain 93 were cultured in HMI9 medium supplemented with 10% FCS at 37 °C under an atmosphere of 5% CO_2_. In all experiments, log-phage cell cultures were harvested by centrifugation at 3000× *g* and immediately used. Drug assays were based on the conversion of a redox-sensitive dye (resazurin) to a fluorescent product by viable cells and were performed according to the manufacturer recommendations (AlamarBlue^®^ Assay, Invitrogen Corporation, Carlsbad, CA, USA) [59,60]. Drug stock solutions were prepared in pure DMSO. *T. brucei brucei* bloodstream forms (3 × 10^4^ cells/mL) were cultured as described above in 96-well plates (200 μL per well) either in the absence or in the presence of different concentrations of inhibitors, with a final DMSO concentration that did not exceed 1%. After a 72 h incubation, resazurin solution was added to each well at with a final concentration of 45 μM. Fluorescence was measured at 530 nm excitation and 590 nm emission wavelengths after a further 4 h incubation. The percentage of inhibition of the parasite growth rate was calculated by comparing the fluorescence of parasites maintained in the presence of drug with that in the absence of drug. DMSO was used as control. Pentamidine was used as an anti-trypanosomal drug control (IC_50_ = 0.011 ± 0.0017 μM). IC_50_s were determined from the dose–response curves with drug concentrations ranging from 100 μM to 50 nM. The IC_50_ value is the mean ± the standard deviation of three independent experiments. Pentamidine was used as an anti-trypanosomal drug control (IC_50_ = 0.011 ± 0.0017 μM).

##### In Vitro Growth Inhibition of Chloroquine-Resistant *P. falciparum*

The FcB1/Colombian strain of *Plasmodium falciparum* was maintained in vitro in human erythrocytes in an RPMI 1640 reaction mixture supplemented with 8% (*v/v*) heat-inactivated human serum at 37 °C under an atmosphere of 3% CO_2_, 6% O_2_, and 91% N_2_. The chloroquine-resistant strain FcB1/Colombia of *Plasmodium falciparum* was obtained from the National Museum Natural History collection, Paris, France (n°MNHN-CEU-224-PfFCB1). Parasites were maintained in vitro on human A+ erythrocytes in RPMI 1640 medium supplemented by 8% (*v/v*) heat-inactivated human serum at 37 °C under an atmosphere of 3% CO_2_, 6% O_2_, and 91% N_2_. Human red blood cells and serum were provided by the Etablissement Français du Sang, Ile de France (EFS, Paris, France) under the C-CPSL-UNT approval n°13/EFS/126. In vitro drug susceptibility assays were measured by [^3^H]-hypoxanthine incorporation as described previously [61]. Drug solutions were prepared in DMSO at a 10 mM concentration. Compounds were serially diluted 2-fold with 100 μL culture medium in 96-well plates. Asynchronous parasite cultures (100 μL, 1% parasitemia and 1% final hematocrit) were then added to each well and incubated for 24 h at 37 °C prior to the addition of 0.5 μCi of [^3^H]-hypoxanthine (GE Healthcare, Tremblay-en-France, France, 1 to 5 Ci·mmol/mL) per well. After a further incubation of 24 h, plates were frozen and thawed. Cell lysates were then collected onto fiberglass filters and counted in a liquid scintillation spectrometer. The growth inhibition for each drug concentration was determined by comparison of the radioactivity incorporated in the treated culture with that in the control culture maintained on the same plate. The concentration causing 50% growth inhibition (IC_50_) was obtained from the drug concentration–response curve, and the results are expressed as the mean values ± standard deviations determined from several independent experiments. Chloroquine was used as an antimalarial drug control (IC_50_ = 0.072 ± 0.0074 μM).

#### 3.9.3. Cytotoxic Assay

Human umbilical vein endothelial cells (HUVECs) were obtained from Clonetics (Lonza; Walkersville, MD, USA) and cultured according to the supplier’s instructions. Briefly, HUVECs from three to six passages were subcultured to confluence onto 0.2% gelatin-coated tissue culture flasks in endothelial cell growth medium (EGM2) containing growth factors and 2% FCS. All cell lines were maintained at 37 °C in a humidified atmosphere containing 5% CO_2_. Cell viability was assessed using Promega (Promega France, Charbonnière-les-Bains, France) CellTiter-BlueTM reagent according to the manufacturer’s instructions. Cells were seeded in 96-well plates (5 × 103 cells per well) containing 50 mL growth medium. After 24 h of culture, the cells were supplemented with 50 mL of test compound dissolved in DMSO (<0.1% in each preparation). After incubation for 72 h, 20 mL resazurin was added for 2 h before recording fluorescence (λex = 560 nm, λem = 590 nm) using a Victor microtiter plate fluorimeter (PerkinElmer France, Villebon-sur-Yvette, France). IC_50_ values correspond to the concentration of test compound that elicited a 50% decrease in fluorescence for drug-treated cells relative to untreated cells. Experiments were performed in triplicate. Fumagillin was used as a positive control and showed an IC_50_ of 0.006 ± 0.002 μM.

## 4. Conclusions

This study combined genomic and metabolomic approaches to study metabolome produced by mutualistic fungal strains of termites from French Guiana of *Pseudallescheria* genus. It allowed the annotation of tyroscherin analogs previously described, as well as pseurotin A, but also new ovalicin analogs.

Our *P. boydii* strain collection is a source of diverse ovalicin derivatives, including new fatty acid esters of ovalicin and boyden C that have been isolated and structurally characterized. Compounds **2**–**4**, similar to ovalicin, exhibited antiplasmodial and antitrypanozomal properties. Our work also confirms that *P. boydii* strains are associated with termite biosynthesized compounds with biological activities. We were able to demonstrate that antiparasitic compounds such as Compounds **1** to **4** are produced in different amounts by the different strains of *P. boydii*. These same compounds reduce cytotoxicities, with IC_50_s greater than 10 µM on HUVECs.

The molecular networking analysis allowed the annotation of all isolated compounds and many other ovalicin analogues, together with the isolation and identification of the new ovalicin analog boyden C. Finally, it was possible by genome mining to propose a common biosynthetic pathway for the ovalicin analogs compounds produced by *Pseudallescheria* sp. strains of our collection. The biosynthesis also permitted the deduction of absolute configurations of ovalicin analogs.

Fungal natural products possess biological activities that are of great value for medicine, agriculture, and manufacturing. Recent “omics” studies accentuated the immense taxonomic diversity of fungi, and the accompanying specialized metabolic diversity provides a significant and still largely untapped resource for natural product discovery. Particular attention will be given in the future to integrative pan-genomic approaches that use a combination of genomic and metabolomic data for parallelized natural product discovery across multiple strains. Such novel technologies will not only expedite the natural product discovery process but will also allow the assembly of a high-quality toolbox for the re-design or even de novo design of biosynthetic pathways using synthetic biology approaches, while continuing to focus on specific environments such as symbiotic microorganisms.

## Figures and Tables

**Figure 1 molecules-27-01182-f001:**
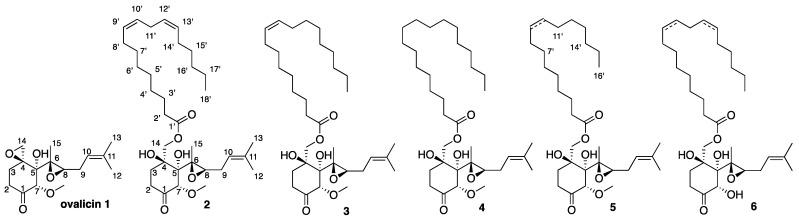
Structure of Compounds **2**–**6** isolated from *P. boydii* SNB-CN85.

**Figure 2 molecules-27-01182-f002:**
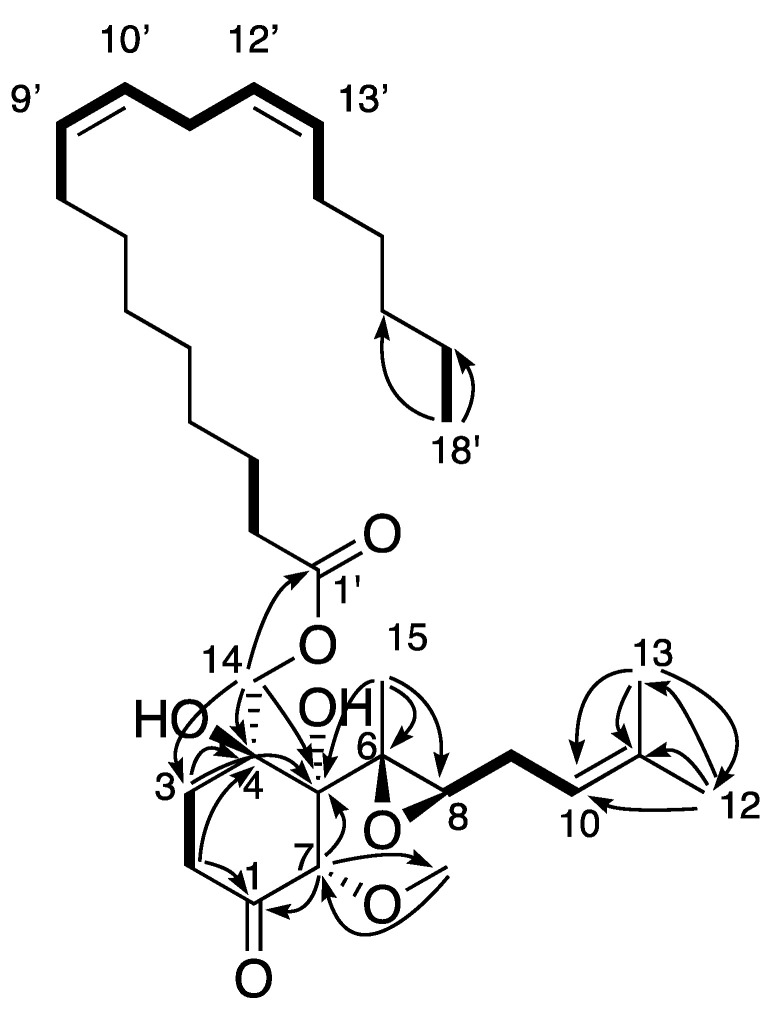
Example of observed key ^1^H-^1^H COSY (bold bonds) and ^1^H-^13^C (plain arrow) correlations observed for **2**–**5** (example from molecule **2**).

**Figure 3 molecules-27-01182-f003:**
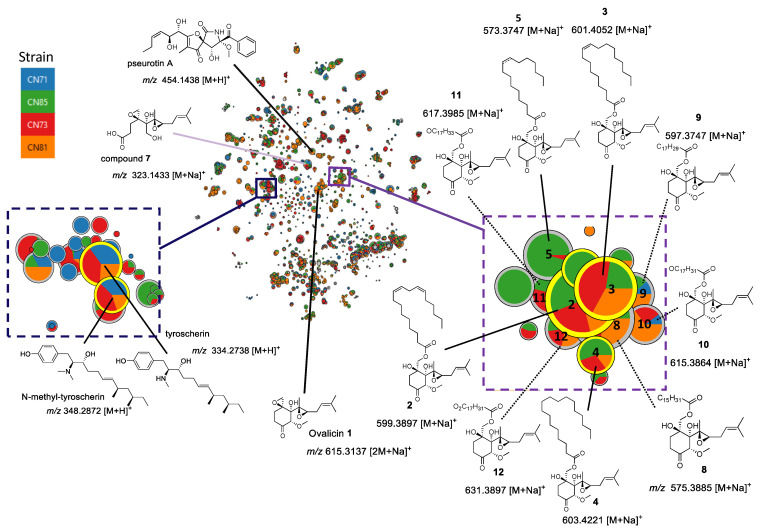
Molecular network with t-SNE representation of crude extracts of *P. boydii*. SNB-CN71, -CN73, -CN81, and -CN85; in purple cluster close to the sodium-cationized ovalicin (**1**), ester analogs and other ovalicin fatty acid derivatives identified by dereplication (dotted lines), in the dark blue cluster of the protonated tyroscherin and *N*-methyltyrocherin.

**Figure 4 molecules-27-01182-f004:**
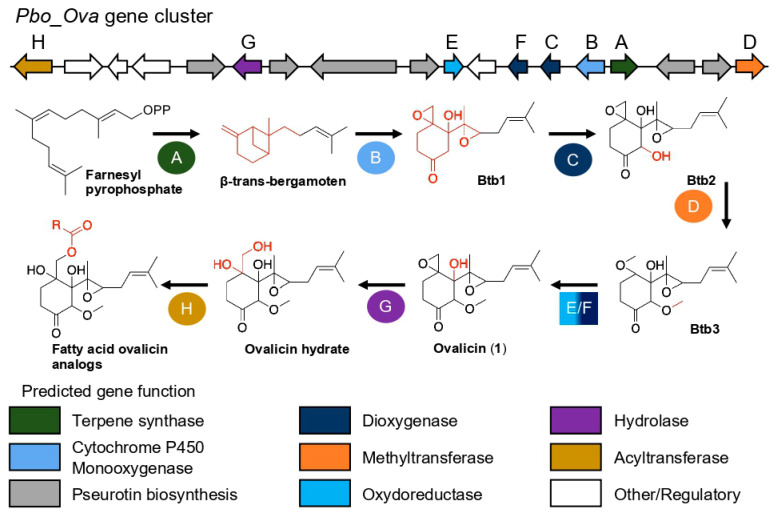
Annotated biosynthetic gene cluster *Pbo_Ova* of *P. boydii* and related biosynthetic pathways (the different analogs of β–trans–bergamoten are indicated under the acronym Btb). R: carbon chain.

**Figure 5 molecules-27-01182-f005:**
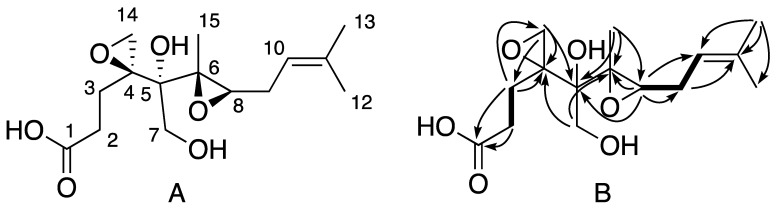
(**A**) Structure of Compound **7** isolated from *P. boydii* SNB-CN73 and (**B**) key ^1^H-^1^H COSY (bold bonds) and long-range ^1^H-^13^C (plain arrow) correlations.

**Table 1 molecules-27-01182-t001:** NMR spectroscopic data of **2**–**6** in CD_3_OD.

N°	2		3		4		5	6	
Pos.	δc, Type	δ_H_ (*J* in Hz)	δc	δ_H_	δc	δ_H_	δ_H_	δc	δ_H_
1	211.1, C		211.1, C		211.2, C			211.6, C	
2	36.5, CH_2_	2.74, m	36.5, CH_2_	2.74, bsex (7.3)	36.5, CH_2_	2.74, td (13.7; 7.6)	2.74, m	36.5, CH_2_	2.76, m
		2.22, m		2.22, m		2.23, m	2.22, m		2.29, m
3	32.8, CH_2_	2.10, dd (13.6, 5.1)	32.7, CH_2_	2.09, m	32.8, CH_2_	2.09, td (13.6; 7.1)	2.09, m	32.8, CH_2_	2.12, m
		2.01, ddd (13.6, 7.1, 2.0)		2.01, m		2.01, ddd (13.6; 7.1; 1.9)	2.01, m		2.04, m
4	76.5, C		76.5, C		76.5 *, C			76.3 *, C	
5	82.8, C		82.7, C		82.8, C			81.8 *, C	
6	62.8, C		62.7 *, C		62.7 *, C			62.3 *, C	
7	86.2, CH	4.69, s	86.2, CH	4.69, s	86.2, CH	4.69, s	4.69, s	77.5, CH	5.02, s
8	58.4, CH	2.97, t (6.5)	58.3, CH	2.97, t (6.5)	58.4, CH	2.97, t (6.5)	2.97, t (6.5)	57.9 *, CH	2.96, t (6.8)
9	28.4, CH_2_	2.38, m	28.4, CH_2_	2.37, m	28.4, CH_2_	2.38, m	2.37, m	28.4, CH_2_	2.38, m
		2.26, m		2.27, m		2.25, m	2.26, m		2.26, m
10	119.8, CH	5.28, m	119.8, CH	5.27, m	119.8, CH	5.27, m	5.27, m	119.9, CH	5.28, m
11	136.1, C		136.1, C		136.1, C			136.0, C	
12	26.1, CH_3_	1.72, brd (1.2)	26.1, CH_3_	1.73, bd (1.1)	26.0, CH_3_	1.73, bs	1.73, bd (1.2)	26.1, CH_3_	1.72, bs
13	18.2, CH_3_	1.68, brd (0.9)	18.2, CH_3_	1.68, bd (0.8)	18.2, CH_3_	1.68, bs	1.68, bd (0.8)	18.2, CH_3_	1.68, bs
14	69.4, CH_2_	4.17, d (11.2)	69.4, CH_2_	4.17, d (11.1)	69.4, CH_2_	4.17, d (11.1)	4.17, d (11.1)	69.5, CH_2_	4.18, d (11.1)
		4.11, d (11.2)		4.11, d (11.1)		4.11, d (11.1)	4.11, d (11.1)		4.12, d (11.1)
15	16.3, CH_3_	1.54, s	16.3, CH_3_	1.55, s	16.3, CH_3_	1.54, s	1.55, s	16.3, CH_3_	1.61, s
OMe	59.7, CH_3_	3.49, s	59.7, CH_3_	3.49, s	59.7, CH_3_	3.48, s	3.49, s	175.2, C	
1′	175.5, C		175.4, C		175.5, C			35.1, CH_2_	2.38, m
2′	35.1, CH_2_	2.38, t (7.4)	35.1, CH_2_	2.38, m	35.1, CH_2_	2.38, m	2.38, m	26.2, CH_2_	1.64, m
3′	26.2, CH_2_	1.64, m	26.1, CH_2_	1.64, m	26.4, CH_2_	1.25–1.37, m	1.64, m	30.2–30.8, CH_2_	1.27–1.40, m
4′–7′	30.3−30.8, CH_2_	1.28, m–1.40, m	30.2–31.0, CH_2_	1.26–1.40, m	30.3–31.0, CH_2_	1.25–1.37, m	1.25–1.40, m	30.4, CH_2_	1.37, m
8′	28.3, CH_2_	2.07, m	28.3, CH_2_	2.03, m	33.2, CH_2_	1.25–1.37, m	2.03, m	28.4, CH_2_	2.07, m
9′	131.0, CH	5.36, m	130.9, CH	5.35, m	23.9, CH_2_	1.25–1.37, m	5.35, m	131.0, CH	5.36, m
10′	129.3, CH	5.33, m	131.1, CH	5.35, m	14.6, CH_3_	1.25–1.37, m	5.35, m	129.3, CH	5.33, m
11′	26.7, CH_2_	2.78, m	28.3, CH_2_	2.03, m	211.2, C	1.25–1.37, m	2.03, m	26.7, CH_2_	2.77, m
12′	129.2, CH	5.33, m	30.2–31.0, CH_2_	1.26–1.40, m	36.5, CH_2_	1.25–1.37, m	1.26–1.40, m	129.2, CH	5.33, m
13′	131.1, CH	5.36, m	30.2–1.0, CH_2_	1.26–1.40, m		1.25–1.37, m	1.30, m	131.1, CH	5.36, m
14′	28.3, CH_2_	2.07, m	30.2–31.0, CH_2_	1.26–1.40, m	32.8, CH_2_	1.25–1.37, m	1.29, m	28.4, CH_2_	2.07, m
15′	30.3–30.8, CH_2_	1.28, m–1.40, m	30.2–31.0, CH_2_	1.26–1.40, m		1.25–1.37, m	0.90, t (7.0)	30.4, CH_2_	1.37, m
16′	32.8, CH_2_	1.31, m	33.2, CH_2_	1.30, m	76.5 *, C	1.29, m		32.8, CH_2_	1.31, m
17′	23.8, CH_2_	1.34, m	23.9, CH_2_	1.29, m	82.8, C	1.29, m		23.8, CH_2_	1.33, m
18′	14.6, CH_3_	0.91, t (7.0)	14.6, CH_3_	0.90, t (7.0)	62.7 *, C	0.90, t (7.0)		14.6	0.91, t (7.2)

*: deduced from HMBC correlations.

**Table 2 molecules-27-01182-t002:** Evaluation of antiparasitic activities of Compounds **1**–**4.**

Cmpd.	*P. falciparum* (IC_50_ µM)	*T. brucei brucei* (IC_50_ µM)	HUVECs (IC_50_ µM)
**1**	13.6 ± 6.4	0.41 ± 0.17	13 ± 2
**2**	19.8 ± 6.7	1.1 ± 0.5	37 ± 3
**3**	>50	4.8 ± 2.7	45 ± 2
**4**	20.6 ± 4.6	4.1 ± 2.1	70 ± 1

**Table 3 molecules-27-01182-t003:** ^1^H and ^13^C NMR data of **7** recorded in CD_3_OD.

Position	*δ*_C_, Type	*δ*_H_ (*J* in Hz)
1	178.0, C	
2	29.3, CH_2_	2.22
3	25.6, CH_2_	2.27, m
		2.05, m
4	61.5, C	
5	76.0, C	
6	63.4, C	
7	63.6, CH_2_	3.64, d (11.6)
		3.60, d (11.6)
8	58.3, CH	3.26, t (6.5)
9	28.0, CH_2_	2.36, m
		2.22, m
10	119.8, C	5.24, m
11	135.4, C	
12	25.8, CH_3_	1.67, s
13	17.9, CH_3_	1.73, s
14	48.8	3.00, d (3.9)
		2.68, d (3.9)
15	17.6, CH_3_	1.37, s

## Data Availability

Not applicable.

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
