# Peer review of "Antiparasitic Ovalicin Derivatives from Pseudallescheria boydii, a Mutualistic Fungus of French Guiana Termites"

_molecules, 2022, doi:10.3390/molecules27041182_

Round 1

Reviewer 1 Report

The authors carried out the modifications proposed by this reviewer. Therefore, the paper could be accepted for publication.

Author Response

Thank you for your comment and for reading our article carefully.

Reviewer 2 Report

The manuscript entitled "Bioactive fatty acid-derived ovalicin from ..." by Jonathan Sorres at all. it's a pretty good piece of work, only needs a few tweaks in my opinion, and could be ready for publication in Molecules. It seems to be quite a valuable job. So apart from mostly technical remarks, it's hard to find any bigger sins in it.
However, I would like to see the work in the track changes mode before possible publication.
Below are my comments.
1. Something could be added in this title to be more informative.
2. The abstract could be slightly supplemented, slightly more precise about the purpose of the work towards its end.
3. The introduction is far too short and needs to be improved in the sense of extended, at least twice. Even by entering only the keywords proposed by the authors of the work, we have a lot of works from which you can really choose a lot.
4. Fig 2 - these arrows are illegible - this part of the molecule should be greatly enlarged. A lot.
5. Fig 5 - panel B - the same, please enlarge this panel somehow to make it more readable.
6. Conclusion - especially the last paragraph should be improved.
In conclusion, I like the work a lot, and it is worth publishing after literally minor corrections. However, the admission needs to be greatly improved as for now it is too weak.
I recommend major revision.

Author Response

We thank you for reviewing our article and we appreciate your remarks allowing an improvement of our work. At your request the follow-up of the corrections has been done.

Our responses are also described below.

The manuscript entitled "Bioactive fatty acid-derived ovalicin from ..." by Jonathan Sorres at all. it's a pretty good piece of work, only needs a few tweaks in my opinion, and could be ready for publication in Molecules. It seems to be quite a valuable job. So apart from mostly technical remarks, it's hard to find any bigger sins in it.
However, I would like to see the work in the track changes mode before possible publication.
Below are my comments.

  1. Something could be added in this title to be more informative.

The title has been changed to be more informative. We hope that it will meet your expectations.

  1. The abstract could be slightly supplemented, slightly more precise about the purpose of the work towards its end.

The abstract was completed mainly by implementing the final data corresponding to the genomic and metabolomic analyses performed.

  1. The introduction is far too short and needs to be improved in the sense of extended, at least twice. Even by entering only the keywords proposed by the authors of the work, we have a lot of works from which you can really choose a lot.

We have expanded the introduction and chosen to add arguments about the choice and rationale for studying the metabolism of termite-associated strains. An addition has also been made concerning the methodology used to study in parallel several strains and perform dereplication by a metabolomic study.

  1. Fig 2 - these arrows are illegible - this part of the molecule should be greatly enlarged. A lot.
    5. Fig 5 - panel B - the same, please enlarge this panel somehow to make it more readable.

We thank you for these remarks. For this purpose, these two figures have been enlarged.

  1. Conclusion - especially the last paragraph should be improved.

We have tried to express in the conclusion a more global view of the combined metabolomics and genomics studies of symbiotic strains. We hope that this will meet your expectations.

In conclusion, I like the work a lot, and it is worth publishing after literally minor corrections. However, the admission needs to be greatly improved as for now it is too weak.
I recommend major revision.

Round 2

Reviewer 2 Report

The authors improved the work very nicely. Now it looks as it should be. Maybe I do not agree with everything, but this does not affect the originality and quality of the work.
I recommend the publication to be acceptable.

This manuscript is a resubmission of an earlier submission. The following is a list of the peer review reports and author responses from that submission.

Round 1

Reviewer 1 Report

The paper describes new fatty acid-ovalicin derivatives of the fungus extract obtained from termites. Biological activities of isolated compounds were investigated as well as the presence of BCG in the genome was annotated. Additionally, a molecular network approach was employed to investigate the presence of ovalicin derivatives in four strains of the fungus. However, some corrections are suggested below to the authors.

The tables describing the NMR data of compounds 2-6 should be grouped and transferred from Supplementary material to the paper.

The epoxy group was represented as β-oriented in the S.M. However, the same group was shown as α-oriented in the paper.  What is the correct stereochemistry of the epoxy group in structures?

The configuration of compounds 2-4 was determined by comparison of the CD of compound 1. Why the absolute stereochemistry of compounds was not carried out by authors through a VCD analysis?  From my point of view, the absolute configuration was not determined for the compounds.

In line 69, the methyl group was lost in compound 6. Please correct the sentence.

Line 83: 1H-1H COSY correlations confirmed the presence of two double bonds for 2 and 6, instead of 2 and 5.

Line 89: .... and fragmentation of in-source acetonitrile adducts.

Figure S44. In the purple chart, the data about ovalicin was lost.  Please, verify this scheme.

The name of microorganisms and parasites should be written in italics. Please revise it in the full text.

Figure 3 should be improved in its presentation. For example, the pie charts are overlaid making it difficult to see the described results.

Author Response

Reviewer 1

The paper describes new fatty acid-ovalicin derivatives of the fungus extract obtained from termites. Biological activities of isolated compounds were investigated as well as the presence of BCG in the genome was annotated. Additionally, a molecular network approach was employed to investigate the presence of ovalicin derivatives in four strains of the fungus. However, some corrections are suggested below to the authors.

The tables describing the NMR data of compounds 2-6 should be grouped and transferred from Supplementary material to the paper.

As you suggestad, Table 1, named “NMR spectroscopic data” regrouping all the NMR 1H and 13C of compounds 2-6 was include to the paper. However, the tables present in the SI have been kept in order to indicate the 2D NMR correlations

The epoxy group was represented as β-oriented in the S.M. However, the same group was shown as α-oriented in the paper.  What is the correct stereochemistry of the epoxy group in structures?

Thank you for this observation, it is an error on our part, we have reviewed attentively the paper where the stereochemistry was incorrect.

The configuration of compounds 2-4 was determined by comparison of the CD of compound 1. Why the absolute stereochemistry of compounds was not carried out by authors through a VCD analysis?  From my point of view, the absolute configuration was not determined for the compounds.

Unfortunately we couldn’t have access a VCD machine. Indeed, the COVID epidemy did not allow us to carry out a collaboration for this type of work.

The configuration proposed is mainly based on the determination of the biosynthetic pathway and is consistent with experimental CD analyses, i.e. the formation of esters from a single precursor: the ovalicin. This scheme allows to put in the relative configurations of this subunit in each of the described compounds. In order to clarify this information, a precision and a sentence has been added line 203.

In line 69, the methyl group was lost in compound 6. Please correct the sentence.

It is indeed compound 6 that has no methyl

Line 83: 1H-1H COSY correlations confirmed the presence of two double bonds for 2 and 6, instead of 2 and 5.

Indeed, it was a mistake

Line 89: .... and fragmentation of in-source acetonitrile adducts.

Done

Figure S44. In the purple chart, the data about ovalicin was lost.  Please, verify this scheme.

The scheme was verified and corrected

The name of microorganisms and parasites should be written in italics. Please revise it in the full text.

As requested, we have checked the whole manuscript taking into account your remark.

Figure 3 should be improved in its presentation. For example, the pie charts are overlaid making it difficult to see the described results.

We have modified the Figure 3, we hope it will be easier to read in this new version.

Reviewer 2 Report

The structural identification of the compounds is not detailed enough, and there are some problems.

Line 69, “For Compound 5, the same chemical shift……” should correct to “For compound 6, ……”

Line 83, “confirmed the presence of two double bonds for 2 and 5, one double bond for 3 and 5, and none for 4”, please revise it.

Line 85, “The Z-configurations of the double bonds were determined by 13C chemical shifts of the double bonds adjacent to methyl groups in accordance with the data from the literature [22,20].” This approach is unreliable and unconvincing.

Line 91, The absolute configurations of Compounds 24 were determined by comparison with circular dichroism data of 1 (Figure S33). Furthermore, all these compounds showed negative optical rotation. Compounds 24 have five chiral centers. Using this approach to determine the absolute configurations, the results are unreliable and unconvincing.

Figure S4, figure S11, figure S25, how to explain the baseline abnormality?

Author Response

Reviewer 2

The structural identification of the compounds is not detailed enough, and there are some problems.

In the results, the compound identification part has been re-read and corrected to clarify the latter

Line 69, “For Compound 5, the same chemical shift……” should correct to “For compound 6, ……”

Line 83, “confirmed the presence of two double bonds for 2 and 5, one double bond for 3 and 5, and none for 4”, please revise it.

Indeed, we have reversed the number of molecules, we have corrected and verified these data in the paper.

Line 85, “The Z-configurations of the double bonds were determined by 13C chemical shifts of the double bonds adjacent to methyl groups in accordance with the data from the literature [22,20].” This approach is unreliable and unconvincing.

In order to justify the stereochemistry of the double bonds, we have based on the following analyses:

1) we obtained identical spectra until 2 and his hemisynthesic compound: 2b with a fatty chain (9cis,12cis), but also the NMR perfect concordances with 3 and 4 and their hemisynthetic compounds.

2) Gunstone's publication which looked at the 13C of the different isomers where we observe a difference on the 13C of methylene at position C-11’ of the chain depending on whether the double bounds are (9cis, 12cis), (9trans,12trans), (9cis,12trans and 9trans, 12trans). A table placed in SI (Table S6) summarizes and illustrates our concordant values.

In order to clarify this information, a precision and a sentence has been added line 90.

Line 91, The absolute configurations of Compounds 24 were determined by comparison with circular dichroism data of 1 (Figure S33). Furthermore, all these compounds showed negative optical rotation. Compounds 24 have five chiral centers. Using this approach to determine the absolute configurations, the results are unreliable and unconvincing.

As indicated in the response to reviewer 1: “The configuration proposed is mainly based on the determination of the biosynthetic pathway and is consistent with experimental CD analyses, i.e. the formation of esters from a single precursor: the ovalicin. This scheme allows to put in the relative configurations of this subunit in each of the described compounds”, a precision and a sentence has been added line  and 203.

Figure S4, figure S11, figure S25, how to explain the baseline abnormality?

The baseline abnormally is unfortunately due for 13C to the high field of our NMR spectrometer, the probe and the sequence. For 13C at the moment we do not have access to the instrument and the software to be able to smooth the baseline.

Round 2

Reviewer 1 Report

Authors carried out all modifications proposed by this reviewer excepted the absolute configuration of compounds. 

Analyzing the supplementary material, I saw the weird baseline of some spectra.  Baseline can be improved with proper spectrum processing.

Author Response

Authors carried out all modifications proposed by this reviewer excepted the absolute configuration of compounds.

Sorry about this. On lines 204-205 in our publication, we wrote '…on subunits' ovalicin relative
configurations' when we should have said absolute configuration since it is indeed the absolute
configuration of the ovalicin that we have determined. We have changed this sentence into ‘…on
subunits' ovalicin absolute configurations.’ In fact, the ovalicin derivatives described in our
publication result from the nucleophilic addition of fatty acids on the terminal position of the
epoxide of ovalicin, and the absolute configuration of ovalicin is obviously conserved. It cannot be
reversed by this process.

Analyzing the supplementary material, I saw the weird baseline of some spectra. Baseline can be improved with
proper spectrum processing.

Sorry again, it is true that the baselines of two 13C spectra were wavy. We unfortunately did not
understand the comment in the first revision, but these baselines have been corrected now.

Reviewer 2 Report

The author did not seriously revise the manuscript.

Figure S4, figure S11, figure S25, the baselines are abnormality. The authors should record the carbon NMR spectrum again.

Author Response

The author did not seriously revise the manuscript.

We worked as hard as we could to do this work and to revise this article.

Figure S4, figure S11, figure S25, the baselines are abnormality. The authors should record the carbon NMR spectrum again.

It is true that the baselines of two 13C spectra were wavy. We unfortunately did not
understand the comment in the first revision.